# ICR-RL: Deep Reinforcement Learning via In-Context Regression

**David Schiff** [1]   **Ofir Lindenbaum** [1]   **Yonathan Efroni** [2]

## Abstract

Recent advancements in machine learning have largely been driven by foundation models (FMs) trained on large, diverse datasets, enabling them to generalize effectively to new, related tasks. However, extending this paradigm to reinforcement learning (RL), where an agent interacts with an environment to select actions, remains a significant challenge. Most existing approaches train FMs directly on sets of control tasks, but developing diverse RL environments and scaling training across them can be costly and complex. In this study, we explore a simpler alternative approach based on a classical reduction from RL to regression. We demonstrate that a foundation model pre-trained for regression tasks, when used as an in-context regression (ICR) model, can be directly applied to RL problems. Building on this insight, we introduce a gradient-free method, `ICR-RL`, that requires no additional training and leverages an ICR foundation model to tackle RL tasks. We evaluate our approach by applying the ICR model with the recently proposed `TabPFN`, which is trained on a wide range of regression tasks. Experiments conducted on the Gymnasium classic-control benchmark indicate that `ICR-RL` can compete with commonly used methods, including DQN, PPO and TRPO. These results show that ICR foundation models can effectively solve RL tasks without fine-tuning, demonstrating their potential as a foundation for RL-oriented models.

## 1. Introduction

Reinforcement Learning (RL) has achieved remarkable success across domains, from game playing, aligning Large Language Models (LLMs) (Ouyang et al., 2022) (Mnih et al., 2013) to robotics (Haarnoja et al., 2024) and amplify-

ing the reasoning capabilities in LLMs (DeepSeek-AI et al., 2025). These advances are primarily driven by gradient-based methods, which exploit the differentiability of neural networks to optimize policies or value functions via stochastic gradient descent (Battash et al., 2024). While effective at enabling end-to-end learning and long-term credit assignment, such methods have their own limitations.

Commonly used deep RL methods impose significant burdens in terms of hyperparameter sensitivity and computational resource allocation. Algorithms such as PPO (Schulman et al., 2017b), TRPO (Schulman et al., 2017a), and DQN (Mnih et al., 2013) are brittle; performance often hinges on the precise tuning of auxiliary parameters, such as clipping ratios, entropy coefficients, and target network update frequencies, and slight misconfigurations can lead to divergences. Furthermore, reliance on backpropagation introduces substantial memory and computational overhead, along with additional hyper-parameters that must be tuned. Unlike inference, training requires storing the full computational graph and intermediate activations to compute gradients, which scales poorly with network depth and sequence length. Ideally, these limitations would be bypassed by a pre-trained Foundation Model (FM) capable of zero-shot or few-shot adaptation, effectively removing the need for extensive per-task backpropagation and hyperparameter search. Yet, constructing such a model poses unique challenges absent in other domains. The primary bottleneck is data diversity: achieving true universality would necessitate training across a massive, heterogeneous corpus of environments. It remains unclear how to curate such a set of datasets and how to train a FM on them.

Recent progress in Large Language Models (LLMs) has demonstrated that transformers can perform a wide range of tasks through In-Context Learning (ICL) (Garg et al., 2023), eliminating the need for gradient-based training or fine-tuning. In this work, we investigate the application of ICL to reinforcement learning by a reduction from an RL task to an In-Context Regression (ICR) task. Intuitively, we treat the ICR model as a regression *oracle*: a black-box mechanism that takes a context dataset of feature-target pairs and a query feature vector, and outputs a target prediction for the query without performing weight updates. We instantiate this oracle using `TabPFN`, a transformer pre-trained on synthetic datasets, and repurpose it as a deep

[1]Bar-Ilan University, Ramat Gan, Israel [2]Tel Aviv University, Tel Aviv, Israel. Correspondence to: David Schiff <davidschiff100@gmail.com>.

*Proceedings of the 43rd International Conference on Machine Learning*, Seoul, South Korea. PMLR 306, 2026. Copyright 2026 by the author(s).

Q-function approximator. This approach establishes a novel gradient-free deep RL framework that can serve as a building block for a future RL FM, in which policy improvement relies solely on the oracle's capacity to regress optimal value estimates directly from the context.

Our results demonstrate that `ICR-RL` can be effectively applied to RL and is competitive with widely used RL methods, such as DQN (Mnih et al., 2013), TRPO (Schulman et al., 2017a), and PPO (Schulman et al., 2017b), in Gymnasium's classic control environments (Towers et al., 2024).

Our contributions are as follows: (1) We develop a framework that uses ICR to approximate the $Q(s, a)$ function (2) We empirically evaluate our method on standard Gymnasium environments, demonstrating its effectiveness in RL (3) We present simple context engineering methods for `ICR-RL` and evaluate them empirically (4) We demonstrate the surprising generalization capability of the `TabPFN` architecture for RL tasks. (5) We propose future steps of creating an RL FM based on `ICR-RL`.

**Conflict of Interest Disclosure**   The authors declare that they have no financial conflicts of interest related to this work.

## 2. Preliminaries

**In-Context Regression**   Let $\mathcal{P}(x, y)$ be a joint distribution over inputs $x$ and targets $y$. We define the test loss for a function $f$ as the expected squared error, $L(f) = \mathbb{E}_{(x,y)\sim\mathcal{P}}[(f(x) - y)^2]$. In this general setting, In-Context Regression (ICR) is viewed as a mapping from a dataset $\mathcal{D} = \{(x_i, y_i)\}_{i=1}^N$ and a query input $x_{\text{query}}$ to a prediction $\widehat{y}(\mathcal{D}, x_{\text{query}})$. Ideally, this mapping approximates the optimal function $f^*$ that minimizes the test loss:

$$\widehat{y}(\mathcal{D}, x_{\text{query}}) \approx f^*(x_{\text{query}}) \quad \text{where} \quad f^* \in \arg\min_f L(f)$$

A transformer model $T_\theta$ is said to perform ICL if it minimizes the expected loss over prompts $P = (\mathcal{D}, x_{\text{query}})$ drawn from the distribution:

$$\min_\theta \mathbb{E}_P \left[ \ell \left( T_\theta(P), y_{\text{query}} \right) \right],$$

without updating parameters $\theta$ during inference.

**Reinforcement Learning**   RL formalizes sequential decision-making as a Markov Decision Process (MDP), defined by the tuple $(\mathcal{S}, \mathcal{A}, P, R, \gamma)$. Here, $\mathcal{S}$ is the state space, $\mathcal{A}$ is the action space, $P(s' \mid s, a)$ is the transition probability of moving to state $s'$ from state $s$ by taking action $a$, $R(s, a)$ is the reward function, and $\gamma \in [0, 1)$ is the discount factor.

We denote an episode $\mathcal{E}$ of length $L$ as a sequence of transitions $\mathcal{E} = \{(s_t, a_t, r_t, s'_t, d_t)\}_{t=0}^{L-1}$, where $s_t, a_t, r_t, s'_t$ rep-

resent the state, action, reward, and next state at step $t$, and $d_t \in \{0, 1\}$ indicates episode termination. The total undiscounted return of an episode is defined as $R_\mathcal{E} = \sum_{t=0}^{L-1} r_t$.

The goal of an RL agent is to learn a policy $\pi(a \mid s)$ that maximizes the expected cumulative discounted reward: $\mathbb{E}_\pi \left[ \sum_{t=0}^\infty \gamma^t R(s_t, a_t) \right]$. Approaches for learning an approximation of the optimal policy include Deep Q-Networks (DQN) (Mnih et al., 2013). DQN relies heavily on gradient-based optimization.

**Offline Reinforcement Learning and Fitted Q Iteration**
Offline reinforcement learning (Offline RL) aims to learn optimal policies from a fixed dataset of previously collected interactions, without further access to the environment. A common and effective approach in Offline RL is *Fitted Q Iteration* (FQI). FQI is a batch-mode variant of Q-learning that performs value iteration using supervised regression. Given a dataset of episodes

$$\mathcal{D} = \bigcup_{i=1}^N \mathcal{E}_i,$$

FQI (Antos et al., 2007) initializes a Q-function $Q_0$ and iteratively performs the following update:

$$y_n^{(k)} = r_n + \gamma \max_{a'} Q_{k-1}(s'_n, a') \quad, \tag{1}$$

for each transition $(s_n, a_n, r_n, s'_n)$, where $k$ is the iteration number and $K$ is the total number of iterations. We define

$$Q_k = \arg\min_{f\in\mathcal{F}} \sum_{i=1}^N \left( f(s_n, a_n) - y_n^{(k)} \right)^2, \tag{2}$$

where $\mathcal{F}$ is a function class (e.g., neural networks, decision trees). This iterative procedure approximates the application of the Bellman optimality operator $\mathcal{T}$. Since $\mathcal{T}$ is a contraction mapping in the sup-norm, the sequence of Q-functions is expected to converge to the unique fixed point $Q_\star$, provided the function approximator is sufficiently expressive, and the dataset is representative. Once the algorithm converges to an approximation of $Q_\star$ after $K$ iterations, a near-optimal policy $\pi_K$ is derived by acting greedily with respect to the learned values:

$$\pi_K(s) = \arg\max_{a\in\mathcal{A}} Q_K(s, a). \tag{3}$$

**`TabPFN`: Prior-Data Fitted Networks**   `TabPFN` is a transformer-based architecture designed for fast and flexible inference on tabular data. Unlike other neural network models such as (Yang et al., 2022; Svirsky & Lindenbaum, 2024), which train separate models for each task, `TabPFN` performs ICL by conditioning on a small training dataset and predicting labels for test points in a single forward pass.

It is trained on a synthetic distribution of millions of small tabular classification and regression tasks, which enables it to learn a strong prior over common data-generating processes.

**TabPFN Architecture and Data**   Concretely, `TabPFN` is a transformer $f_\theta$ trained to map an input sequence containing both training examples and unlabeled samples to label predictions. `TabPFN` was pretrained on a large amount of different synthetic classification and regression tasks. We will denote a single such task by $\mathcal{D}_{\text{task}}^t = \{(x_j, y_j)\}_{j=1}^M$ whose rows are *i.i.d.*: once a latent task $t \sim p(t)$ is fixed, every pair $(x_j, y_j)$ is sampled independently from the same distribution $p(x, y \mid t)$. Each task is generated using causal models and Bayesian networks. The set of tasks that comprises the entire pretraining data which `TabPFN` was trained on is denoted by $\mathcal{D}_{\text{pretrain}} = \{\mathcal{D}_{\text{task}}^1, \mathcal{D}_{\text{task}}^2, \mathcal{D}_{\text{task}}^3 \ldots \mathcal{D}_{\text{task}}^T\}$

During pretraining, each training example drawn from a single task, $(x_j, y_j)$ is embedded as a token $p_j = \text{Embed}(x_j, y_j)$, and each unlabeled input $x_{\text{test}}$ is embedded as a token $q = \text{Embed}(x_{\text{test}})$. The full input to the model is then a sequence $S = [p_1, \ldots, p_n, q_1, \ldots, q_m]$. Self-attention layers enable unlabeled tokens $q_k$ to attend to the training tokens $\{p_j\}$, thereby conditioning on the training set and extracting the relevant mapping from inputs to outputs. When pretraining, a synthetic task is drawn from $\mathcal{D}_{\text{pretrain}}$ and split into $\mathcal{D}_{\text{train}}$ and $\mathcal{D}_{\text{test}}$, and the model parameters $\theta$ are optimized by minimizing the cross-entropy loss $\mathcal{L}(\theta) = \sum_{(x,y) \in \mathcal{D}_{\text{test}}} \ell(f_\theta(\mathcal{D}_{\text{train}}, x), y)$, where $\ell$ is the standard classification loss.

**Inference Using `TabPFN`**   Once trained, $f_\theta$ does not require gradient updates or refitting: for a new dataset $\mathcal{D}$ and queries $x_{\text{test}}$, predictions are obtained simply by concatenating training and query tokens into a sequence and running a single forward pass through the transformer, i.e., $\hat{y}_{\text{test}} = f_\theta(\mathcal{D}, x_{\text{test}})$. For regression tasks, `TabPFN` is trained similarly, except that continuous outputs are discretized into buckets, which are aggregated into a single prediction via a weighted average.

## 3. Related Work

**Transformers in Reinforcement Learning.**   Offline-RL formulations, such as the decision transformer, frame decision-making as a conditional sequence modeling problem, and achieve state-of-the-art results on Atari and OpenAI Gym benchmarks without explicitly learning a value function. They condition the model to predict the sequence of actions given states and the future returns. This method uniquely uses transformers for RL but requires backpropagation and, therefore, gradients (Chen et al., 2021).

The trajectory transformer re-frames offline RL as a se-

quence modeling problem, treating entire trajectories as sequences to be modeled by a transformer. Instead of learning value functions, policies, or dynamics models separately, it models the joint distribution over states, actions, rewards, and optionally returns-to-go, and uses beam search to plan. This approach requires training the trajectory transformer on these sequences and therefore necessitates the use of gradients (Janner et al., 2021).

**Gradient-Free Reinforcement Learning.**   Derivative-free policy search has long served as an alternative to back-propagation. Modern evolutionary algorithms scale to thousands of CPU (central processing unit) cores and achieve competitive performance in RL environments such as Atari, despite noisy gradients (Salimans et al., 2017), although these strategies are computationally intensive.

The KNN-TD(Martín H. et al., 2009) Online RL algorithm uses K-Nearest Neighbors to approximate the Q-function by updating the estimated Q-values based on the nearest neighbors. This approach is also computationally intensive and assumes a linear relationship between the state space and the Q-function, i.e., that states and actions with smaller Euclidean distances have more similar Q-values. This assumption is invalid in complex, nonlinear environments and is addressed by employing a more capable function approximation, such as deep neural networks (Martín H. et al., 2009). Our `TabPFN` RL algorithm departs from these by leveraging a meta-learned prior to fit Q-values, eliminating the need for backpropagation or computationally intensive evolutionary search methods.

**In-Context and Meta Reinforcement Learning.**   Meta-RL aims to learn agents that can rapidly adapt to new tasks. To the best of our knowledge, $RL^2$(Duan et al., 2016) is the first work to show how ICRL works in practice. $RL^2$ uses a GRU (Gated Recurrent Unit) based network (Chung et al., 2014) and Trust Region Policy Optimization (TRPO) (Schulman et al., 2017a) to learn the "learning algorithm" needed to solve a particular distribution of environments. This "slow" learning stage encodes the learned RL algorithm in the GRU's activations. Once this slow meta-training phase is completed, the learned algorithm can be deployed to new environments drawn from the same training distribution, without further gradient updates. In contrast, our method requires no RL-specific training at any stage: TabPFN is trained on generic regression tasks from a different data distribution, yet generalizes zero-shot to RL and achieves performance competitive with state-of-the-art RL algorithms

Algorithm Distillation showed that a transformer can learn an RL algorithm. Given a context of past transitions and the learning history of an RL algorithm, such as DQN, it can produce actions that mimic the behavior of other RL algorithms. The transformer can then learn to improve its

policy in a new unseen environment without any gradient updates (Laskin et al., 2022). Like RL², AD is limited to the environments on which it was trained, whereas our method generalizes to completely OOD environments.

Another method, called ICQL (Xu et al., 2025), uses ICL for offline reinforcement learning but requires gradient-based training for each environment, whereas our method works out of the box on unseen environments using ICL only.

Self-supervised pre-training has recently emerged as a path toward Behavioral Foundation Models (BFMs), in which agents learn a versatile latent space of skills from reward-free data, enabling them to solve diverse downstream tasks by retrieving behaviors rather than learning from scratch. (Sikchi et al., 2025) demonstrates that while simple zero-shot retrieval is often suboptimal, this pre-trained latent space contains high-performance policies that can be accessed using optimization. This approach frames the FM as a specialized RL architecture requiring domain-specific pre-training and test-time parameter updates. In contrast, our `ICR-RL` framework establishes a reduction from RL to generic in-context regression, using a general-purpose oracle such as `TabPFN` rather than a domain-specific RL model. This enables adaptation via inference alone, eliminating the need for gradient computation or specialized latent optimization.

Another related work proposes a transformer-based in-context RL agent trained via large-scale meta-learning across procedurally generated tasks. The training regime uses an artificial environment generator to produce a vast, diverse set of strictly discrete MDPs with randomized transition and reward structures. Rather than learning a single policy, OmniRL (Wang et al., 2025) is trained to model causal sequences of interaction histories across different types of policies and to adapt behavior solely by conditioning on context. While effective, this approach fundamentally depends on massive, domain-specific pre-training on RL trajectories to internalize the structure of MDPs. In contrast, our `ICR-RL` framework removes the need for any RL-specific pre-training. By reducing reinforcement learning to an in-context regression problem, we directly reuse a general-purpose regression FM (e.g., `TabPFN`) that has never been exposed to MDP dynamics or RL trajectories. Moreover, whereas OmniRL is restricted to discrete environments, our regression-based reduction naturally supports both discrete and continuous state–action spaces, providing a more general and flexible foundation for gradient-free deep RL. See Table 1 in Appendix Section A for a comparison of the different in-context RL methods and their properties.

# 4. Methods: `ICR-FQI` and `ICR-RL`

FQI enables a natural reduction from reinforcement learning to in-context regression. To evaluate the effectiveness of this reduction, we instantiate our framework using `TabPFN`, an existing ICR FM. While we use `TabPFN` as the backbone for `ICR-RL`, the proposed approach is not tied to any specific ICR.

To clearly distinguish the components of our framework, we define `ICR-FQI` as the core inference mechanism used to estimate value functions via regression targets, and `ICR-RL` as the overarching online reinforcement learning agent that utilizes this mechanism for control, data collection, and context management.

**In-Context Fitted Q-Iteration (`ICR-FQI`).** As illustrated in the left panel of Figure 1, we employ `ICR-FQI` to estimate target Q-values by approximating $Q^*$. Unlike standard FQI, which updates network weights, `ICR-FQI` is an inference-only procedure. Given a set of environment interactions $\mathcal{D}_0 = \{(s_1, a_1, r_1)...(s_n, a_n, r_n)\}$. Both the ICR context and query are populated by $\mathcal{D}_0$. We iteratively update the target values within the context by computing Bellman targets based on the model's own predictions from the previous iteration, effectively "refitting" the data by running feedforward on the context and query. In each iteration, we run the feedforward and update the context with the new data. After each iteration the value estimation for each state and action becomes more accurate such that $\mathcal{D}_k = \{(s_1, a_1, r_1 + \gamma V_k(s'_1), ...(s_n, a_n, r_n + \gamma V_k(s'_N)\}$ and accordingly, if we run `ICR-FQI` a sufficient number of iterations, and under proper data coverage assumptions, $V_k$ converges to $V^*$ (Antos et al., 2007).

**Online Interaction and Inference (`ICR-RL`).** Following initialization, we transition to an online learning phase (Figure 1, right). The agent interacts with the environment using an $\varepsilon$-greedy policy with $\varepsilon$ scheduling derived from the `TabPFN` Q-network. At every step $t$, given the current state $s_{\text{curr}}$ we construct a query batch consisting of $s_{\text{curr}}$ paired with every possible action $a \in \mathcal{A}$. The model processes the full context $\mathcal{C}$, and these queries in a single forward pass to output $Q(s_{\text{curr}}, \cdot)$, from which the greedy action is selected. The exploration rate $\varepsilon$ decays over episodes, shifting from exploration to exploitation.

**Context Engineering** To ensure the context remains within the "token" budget for runtime reduction, we filter incoming episodes. Let $\mathcal{E}$ be a newly collected episode. We maintain a history of returns $\mathcal{H}$ from episodes currently in the context. A new episode $\mathcal{E}$ is appended to the context only if its return satisfies the predicate $\text{ADD}(\mathcal{E}) = \mathbf{1}\{R_{\mathcal{E}} > \text{Quantile}_{0.95}(\mathcal{H})\}$. We utilize the undiscounted return $R_{\mathcal{E}}$ for this selection criterion to decouple data quality assess-

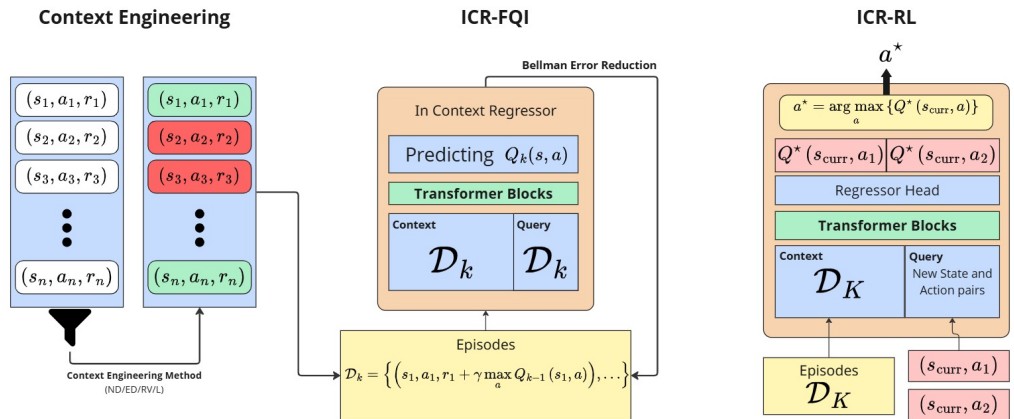

*Figure 1.* The Gradient Free Deep Reinforcement Learning Method. Left: a context engineering method truncates the context to remove any unnecessary tuples $(s_n, a_n, r_n)$ from the context. Middle: the iterative `ICR-FQI` algorithm is shown to reduce the Bellman error at each iteration on the truncated dataset. Right: our `ICR-RL` method selects the best action to take, given a current state $s_{\text{curr}}$ and a context filled with tuples of $(s, a, Q_K(s, a))$, which are the product of $K$ iterations of the `ICR-FQI` algorithm.

---

**Algorithm 1** In-Context Regression RL (`ICR-RL`)

---

**Require:** $B$-context window budget;
$\qquad\qquad \varepsilon - \text{schedule}(k)$-maps iteration to $\varepsilon$ value.
1: $\mathcal{S} \leftarrow \emptyset$           /* *context buffer* */
2: **for** $k = 1, 2, \dots$ **do**
3:     $\varepsilon_k \leftarrow \varepsilon - \text{schedule}(k)$
4:     Collect trajectory $\mathcal{E}_k$ with $\varepsilon_k$-greedy policy over $Q_\theta$
5:     $R_{\mathcal{E}_k} \leftarrow \sum_{(s,a,r) \in \mathcal{E}_k} r$
            /* *Get episode reward and compare* */
6:     **if** $R_{\mathcal{E}_k} > \text{Quantile}_{0.95}(\{R_{\text{hist}}\})$ **then**
7:
8:       **if** $|\mathcal{S}| < B$ **then**
9:         $\mathcal{S} \leftarrow \mathcal{S} \cup \mathcal{E}_k$
10:        $Q_\theta \leftarrow \texttt{ICR-FQI}(\mathcal{S})$
11:       **end if**
12:     **end if**
13: **end for**

---

ment from the optimization horizon. While the discount factor $\gamma$ is necessary for the contraction of the FQI value updates, using it for data selection would arbitrarily penalize long-horizon successes. Filtering by undiscounted returns ensures the context retains trajectories that effectively solve the task, preserving valuable transitions even if they occur late in the episode. This high-reward gate prioritizes trajectories that outperform the historical average, thereby facilitating a self-improving curriculum. Whenever an episode is added, we run `ICR-FQI` to propagate the new reward information throughout the context.

We evaluate `ICR-RL` on three different Gymnasium environments (Towers et al., 2024) (CartPole-v1, MountainCar-v0, Acrobot-v1) and find that `ICR-RL` is competitive with DQN while also outperforming TRPO and PPO. (Mnih et al.,

2013). We show the results of our experiments in Figure 2. During our preliminary experimentation with `ICR-RL`, we observed that `ICR-RL` tends not to perform well with sparse rewards, though further work is required to analyze this effect. Therefore, we employed a reward-shaping method for each environment that forces a continuous reward signal (see Appendix B.2 for details). From a theoretical perspective, the effectiveness of `ICR-RL` in RL is notable. This is because `TabPFN` was previously trained on synthetic data under entirely different assumptions (See Discussion 5 for more details).

### 4.1. Context Engineering for Learning When The Context Budget Is Exhausted

**The Need for Context Engineering** Since transformers are quadratically complex with respect to the length of the context, our method's runtime heavily depends on the size of the context itself. Crucially, the smaller the context, the faster `ICR-RL` will run. In this section, we explore various context engineering methods that enable ICL-based methods to learn with a fixed context budget. Let $B$ denote the maximum number of samples we allow in the context. When using an ICL RL algorithm, more data is collected into the context with each episode, and so two questions arise: (1) Given our limited space, how do we choose the most information-dense samples? (2) When the context is full, how do we continue learning?

**Existing Methods** As discussed in the previous section, we address the first problem by introducing the high reward gate 1. The second problem can be reduced to finding a subset of transitions $\mathcal{S}'$ that can be removed from the context such that the in-context learner still retains similar performance. This challenging problem has been discussed

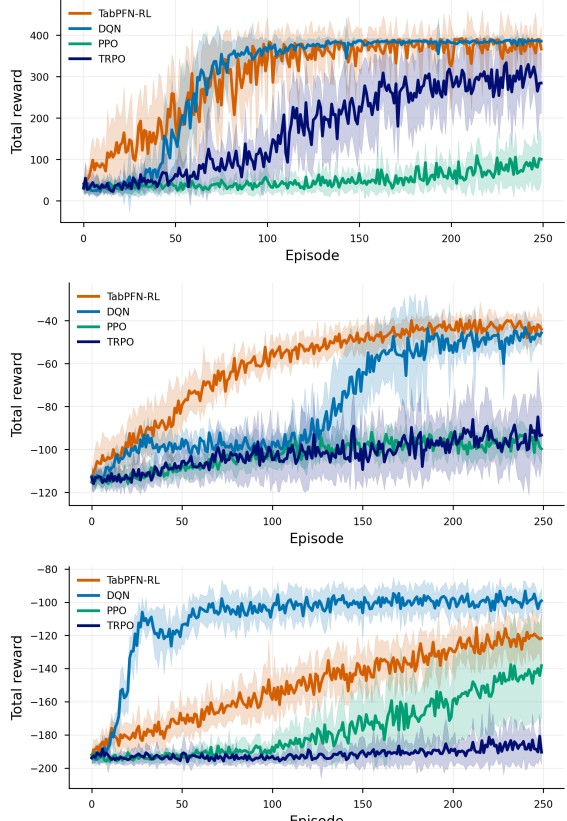

*Figure 2.* Performance on reward per-episode on Cartpole-v1, MountainCar-v0 and Acrobot-v1 respectively (from top to bottom). `ICR-RL` (orange) is compared to DQN (blue), PPO (green), and TRPO (blue).

in (Yang et al., 2025). To the best of our knowledge, no existing ICL method for RL has addressed the question of what to do when the context is fully specified. Some works suggest taking the latest transitions that fit within the budget $B$. We will show that this method has some issues (Wang et al., 2025). The main limitation of this method is that removing older episodes and replacing them with new ones can remove significant contextual information. A simple example of essential information removal from the context is an episode with a high-reward episode, which we would want to keep in the context so that our model can exploit the information about achieving such a high reward. We can view the problem of retaining informative episodes within a fixed context budget while freeing space to add new high-signal episodes as a form of context engineering.

**Context Engineering Methods** We present four approaches to continual learning with context engineering under a fixed budget $B$. We compare these methods against a "Stale Context" ($SC$ in Figure 3) baseline, meaning we run the model without updating the context after $B$ transitions. Another baseline we compare it against is retaining

the latest $B$ transitions. Empirically, we find that our methods outperform baselines and show promise for In-Context RL, which is not limited by context size but rather by the algorithm's context-engineering method for retaining proper transitions.

**Context Engineering Techniques** Let $\mathcal{D}_k$ denote the context (as defined before) with capacity $B$ and let the newly finished episode of length $L$ be $\mathcal{E} = \{(s_t, a_t, r_t)\}_{t=1}^L$. We implement four replacement heuristics for maintaining $\mathcal{D}_k$ once it is full:

LATEST EPISODES (L) This is a pure first-in/first-out strategy that *appends* $\mathcal{E}$ and, if $|\mathcal{D}_k| > B$, discards the oldest $|\mathcal{D}_k| - B$ transitions. It keeps exactly the $B$ most recent transitions and thus performs no value- or diversity-based filtering.

NAIVE DE-DUPLICATION (ND). We build a lower-triangular pairwise distance matrix over all pairs of $x_n = (s_n, \text{onehot}(a_n))$ that are in-context $\mathcal{D}_k$. Such that the distance is $D_{nm} = \|x_n - x_m\|_2$ for all stored transitions, we then identify, for each transition $x_n$, its closest neighbor $\hat{j}(n) = \arg\min_{m<n} D_{nm}$, and mark the transition with the smaller row index in each closest pair for removal. We choose the $L$ transitions $x_n$ with the smallest distance to their neighbor, removing one element from each pair of near-duplicates. Intuitively, we remove duplicate transitions from the context by identifying near-duplicate pairs and retaining only one.

EMBEDDINGS DE-DUPLICATION (ED). Identical logic to the naive variant, but redundancy is computed in the model's representation space. Let $z(\cdot)$ map a transition $x_n$ to its transformer encoder representations. We then run the Naive Deduplication algorithm on $z(x_n)$. This representation-aware pruning preserves a more diverse set of semantics than raw state–action coordinates.

REWARD VARIANCE (RV). We add episodes to the context freely until the budget $B$ is depleted. Once $|\mathcal{D}_k| = B$, we partition the context into a *high-performing* set $\mathcal{G}$ and a *low-performing* set $\mathcal{W}$. Let $\tau$ denote the median return of all episodes currently in $\mathcal{D}_k$. We formally define these partitions such that: $\mathcal{G} = \{\mathcal{E} \in \mathcal{D}_K \mid R(\mathcal{E}) \geq \tau\}$ and $\mathcal{W} = \{\mathcal{E} \in \mathcal{D}_k \mid R(\mathcal{E}) < \tau\}$. When a new episode $\mathcal{E}_{\text{new}}$ arrives while the context is full, we determine inclusion based on the distributions within these partitions: (R1) If $R(\mathcal{E}_{\text{new}})$ exceeds the 95th percentile of returns in $\mathcal{G}$, we evict the lowest-return episode from $\mathcal{G}$ and insert $\mathcal{E}_{\text{new}}$. (R2) If $R(\mathcal{E}_{\text{new}})$ is below the 5th percentile of returns in $\mathcal{W}$, we evict the highest-return episode from $\mathcal{W}$ and insert $\mathcal{E}_{\text{new}}$. (R3) Otherwise, the new episode is discarded. This heuristic promotes a balanced mixture of superior exemplars

and distinct failure modes, generating richer learning signals than recency or geometric diversity alone.

## 4.2. Results

We evaluate `ICR-RL` on three environments from the gymnasium ([Towers et al., 2024](#)) framework. We compare `ICR-RL` to DQN, PPO, and TRPO. We executed each experiment for 250 episodes. Our results show that `ICR-RL` yields competitive performance to DQN while also outperforming PPO and TRPO without requiring backpropagation (See figure [2](#)). Please refer to the technical Appendix for implementation details.

We further demonstrate that our model supports continuous learning, even when the model context budget $B$ is utilized. We demonstrate the effectiveness of different context engineering methods and observe that the ND method yields the best results. We also show that the naive deduplication method allows for continual learning, even after hundreds of episodes and when the context budget is depleted. All other methods show continual improvement, but, as shown in Figure [3](#), naive deduplication (ND) outperforms them. Clearly, using the latest episode is the worst context-engineering strategy for continued learning.

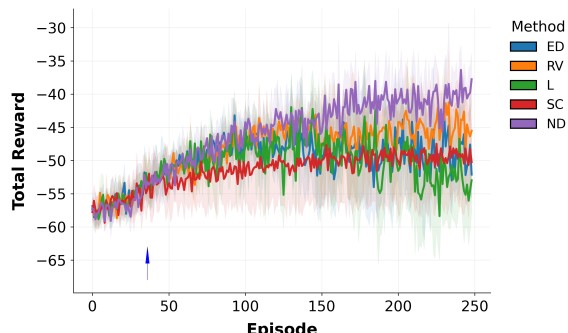

*Figure 3.* Effectiveness of context engineering methods on continued learning after the context is full. The blue arrow points to the latest episode across all experiments in which the context is filled, and the truncation process must start. The best method, in green, Naive De-duplication (ND), performs well and exhibits a strong learning curve. Other methods initially surpass the stale context (SC), but they seem to fail as the episodes progress.

## 4.3. Sparse Rewards

Although our method performed well in environments with smoothed rewards, we found that using a different ICR backbone, such as TabICL ([Qu et al., 2026](#)), significantly improved our performance in sparse-reward environments. This experiment showcases the generality of our framework and shows that different in context regression models which were trained on different priors can be better in different settings.

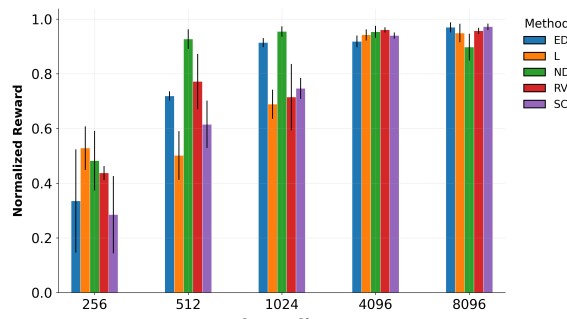

*Figure 4.* Context Engineering methods could cut down memory usage by using a smaller context size while performing competitively with significantly larger context sizes. Note the significant performance achieved using just 512 samples in context, on par with the performance of a context size of 8096.

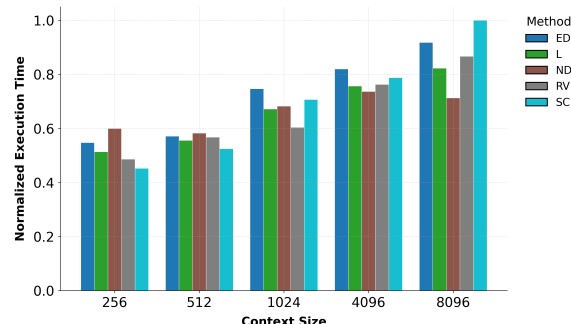

*Figure 5.* Effectiveness of context engineering methods on algorithm runtime. The values are normalized relative to the longest running execution time. Runtime is reduced by nearly 40% when using the proper context engineering method, while maintaining similar performance.

## 4.4. Ablation study

In this section, we focus on the various parameters used by the `ICR-RL` algorithm. We find that different parameter values significantly affect the model's convergence, though having many fewer hyper-parameters means we can spare precious computation time by scanning for the best ones. This is significant, as in classic deep RL, we would need to scan a large space of possible neural network architectures and gradient optimization settings. For each hyperparameter in `ICR-RL`, we evaluated multiple values and assessed their impact on model convergence. Each parameter value is evaluated on five seeds. We conduct all ablation experiments in the MountainCar-v0 environment for 300 episodes, with each episode limited to 200 steps. See Figure [7](#). For more ablation experiments, see appendix [C](#)

## 5. Discussion

**TabPFN and Task-Prior Mismatch** `TabPFN` was trained on synthetic tabular datasets where samples are

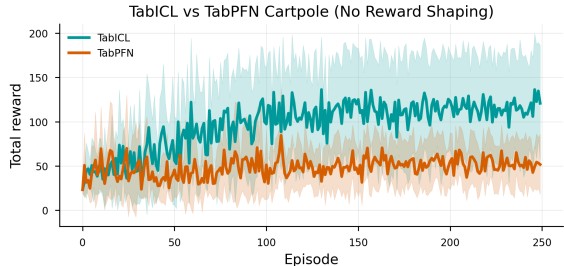

*Figure 6.* TabPFN and TabICL([Qu et al., 2026](#)) on sparse reward Cartpole. Notably, we see that TabICL significantly outperforms TabPFN, showing a strong learning curve. In this experiment, TabPFN and TabICL were compared with the same hyperparameters exactly.

strictly i.i.d. Our application of `TabPFN` as the backbone of `ICR-RL` violates this structural assumption, pushing the model out of distribution. First, RL data violates independence: the context consists of correlated trajectory segments rather than exchangeable samples. Second, the regression targets are bootstrapped and non-stationary. The target $y_n = r_n + \gamma \max_{a'} Q_{k-1}(s'_n, a')$ includes approximation errors from the previous iteration, introducing structured noise that differs fundamentally from the noise seen during pre-training. Third, the data distribution shifts over time as the policy evolves, violating the assumption of a static data-generating process. The fact that `TabPFN` effectively minimizes the Bellman error despite these mismatches highlights an unexpected robustness in in-context regression priors.

## 6. Conclusions and Future Work

In this work, we presented `ICR-RL`, a novel framework that bridges the gap between In-Context Regression and Reinforcement Learning. By reducing the RL problem to a regression task via `ICR-FQI`, we demonstrated that a pre-trained regression FM, specifically `TabPFN` can solve control tasks without any gradient-based training or fine-tuning. Our empirical results on Gymnasium environments show that `ICR-RL` matches or exceeds the performance of established baselines, such as DQN. Furthermore, we introduced novel context-engineering techniques, such as Naive De-duplication, that enable continual learning within a fixed context budget, effectively managing the trade-off between memory constraints and data diversity.

A key conceptual contribution of this work is the validation of the reduction hypothesis: improvements in general-purpose Regression FMs can directly translate to improvements in Reinforcement Learning FM. This suggests that the path to an RL gradient-free FM may not necessarily require training on vast corpora of trajectories, but rather on building a more robust and capable ICR.

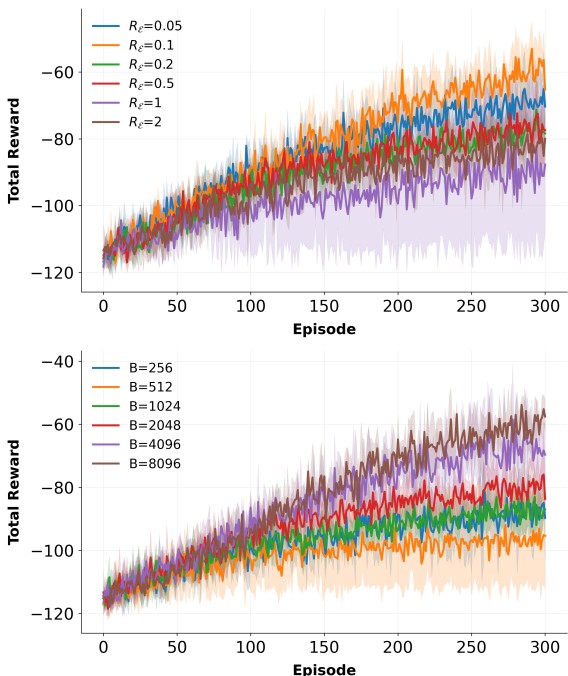

*Figure 7.* Effects of the various parameters of `ICR-RL` on model convergence (from top to bottom). We ablate the percentile refit parameter by trying different values (0.4, 0.6, 0.7, 0.9, 0.95). We observe nearly identical top performance for 0.9 and 0.95. We also evaluate the effect of the context budget $B$ by varying its size from 256 to 8192 in multiples of 8 and find that the best performance is achieved with a size of 8096.

**Future Directions.** We identify several promising avenues for future research to scale `ICR-RL` to a general-purpose agent:

- **Addressing the Prior-Data Mismatch:** As discussed, `TabPFN` is trained on i.i.d. data, whereas FQI generates non-stationary, bootstrapped targets. Future work involves fine-tuning ICR models specifically on datasets that mimic the statistical properties of FQI (e.g., varying noise levels and distribution shifts). Aligning the pre-training prior with the FQI posterior should significantly improve stability.

- **One-Shot Latent FQI:** Currently, `ICR-RL` performs FQI explicitly by iteratively re-labelling the context. A compelling direction is to fine-tune the ICR model to perform these iterations implicitly within its forward pass. By training the model to predict the fixed-point $Q^*$ directly from the initial transition data, we could achieve "One-Shot FQI," effectively moving the optimization loop into the transformer's latent space.

- **Scaling to High-Dimensional Spaces:** Our evaluation focused on classic control tasks with low-dimensional feature spaces. Extending `ICR-RL` to high-dimensional

inputs, such as visual observations or complex robotics state-spaces, remains a critical next step. This may require integrating `TabPFN` with pre-trained visual encoders or scaling the underlying transformer architecture to handle higher feature complexity.

Ultimately, `ICR-RL` serves as a proof-of-concept for a new paradigm in RL: replacing complex, unstable training loops with inference-time adaptation enabled by strong, general-purpose FMs.

## Limitations

`ICR-RL` shows potential, yet limitations remain. Scaling to high-dimensional tasks is non-trivial and has not yet been tested. Additionally, our current reliance on dense reward shaping suggests that our method struggles with sparse rewards relative to specialized exploration methods. Finally, the quadratic runtime complexity of the transformer is a challenge, slowing `ICR-RL` training and inference, but a linear transformer, as in (Zeng et al., 2025), may improve this. Further work is needed to address these issues.

## Impact Statement

This paper presents work aimed at advancing the efficiency and universality of Reinforcement Learning systems. While FMs have transformed supervised learning, extending their capabilities to RL is a non-trivial feat. Our work helps address this barrier by demonstrating that capable control policies can emerge directly from general-purpose regression priors via in-context learning, without requiring gradient-based fine-tuning. This reduction of reinforcement learning to in-context regression paves the way for more sustainable, energy-efficient AI agents and deepens our fundamental understanding of how transformers can generalize to dynamic, non-stationary environments outside their training distribution.

## Acknowledgments

YE is partially supported by the Israeli Science Foundation (ISF) grant no 4032/25. OL was supported by the MOST grant No. 0007341.

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

## A. In Context Learning RL Methods Comparison

To contextualize our approach, Table 1 compares `ICR-RL` with standard Deep RL baselines, transformer-based methods, and meta-RL algorithms. The comparison highlights key operational differences, specifically training requirements, the ability to handle continuous-state spaces, and the necessity of domain-specific pretraining.

| Method | Gradient-Free | Zero-Shot | No RL Pretraining | Continuous States | Online Interaction |
|---|---|---|---|---|---|
| DQN (Mnih et al., 2013) / PPO (Schulman et al., 2017b) / TRPO (Schulman et al., 2017a) | ✗ | ✗ | ✓ | ✓ | ✓ |
| Decision Transformer (Chen et al., 2021) | ✗ | ✗ | ✗ | ✓ | ✗ |
| Trajectory Transformer (Janner et al., 2021) | ✗ | ✗ | ✗ | ✓ | ✗ |
| RL$^2$ (Duan et al., 2016) | ✗ | ✗ | ✗ | ✓ | ✓ |
| Algorithm Distillation (Laskin et al., 2022) | ✗ | ✗ | ✗ | ✓ | ✓ |
| ICQL (Xu et al., 2025) | ✗ | ✗ | ✗ | ✓ | ✗ |
| OmniRL (Wang et al., 2025) | ✓ | ✓ | ✗ | ✗ | ✓ |
| **ICR-RL (ours)** | ✓ | ✓ | ✓ | ✓ | ✓ |

*Table 1.* Comparison of ICR-RL with representative deep RL, transformer-based RL, and in-context/meta-RL methods. ICR-RL uniquely combines gradient-free learning, zero-shot generalization to unseen environments, no RL-specific pretraining, continuous state-action support, online interaction, and explicit context engineering.

## B. Implementation Details

The code for the experiments was implemented in `Python 3.9.21` using the `TabPFN` library (version 2.0.9) and Stable-Baselines3 (version 2.6.0). All experiments were conducted on a single Nvidia A100 GPU server with an Intel(R) Xeon(R) Gold 6338 CPU @ 2.00 GHz. For our comparison with DQN, PPO, and TRPO on the Gymnasium classic control (Towers et al., 2024) environments (Acrobot-v1, MountainCar-v0, and CartPole-v1), we ran 10 seeds per environment.

We ran `ICR-RL` using a multi-phase $\varepsilon$-greedy schedule defined by the parameters $N_{\varepsilon_0}$, $\varepsilon_0$, $\lambda$, and $\varepsilon_{min}$. Here, $N_{\varepsilon_0}$ represents the number of initial episodes where $\varepsilon$ is fixed at 1.0 to ensure maximal initial exploration. For our experiments, we set $N_{\varepsilon_0} = 1$. Immediately following these $N_{\varepsilon_0}$ episodes, $\varepsilon$ is reset to the baseline value $\varepsilon_0$, which is set to 0.95 for Acrobot-v1 and 0.7 for both MountainCar-v0 and CartPole-v1 before decaying. For all subsequent episodes, $\varepsilon$ decays exponentially by $\lambda$ until it reaches the floor $\varepsilon_{min} = 0.1$. The decay factors $\lambda$ were 0.9955 for Acrobot-v1 and 0.99 for the other two environments. For the `ICR-FQI` algorithm, $\alpha$ was set to 0.1 while $\gamma$ was 0.99 for all environments. Before the first `ICR-FQI` iteration, we perturb every reward with an infinitesimal uniform noise, $r \leftarrow r + \varepsilon$, $\varepsilon \sim \mathcal{U}(0, 10^{-4})$. This guarantees a non-zero standard deviation after `TabPFN`'s internal $z$-score normalization. Not adding uniform noise would result in a division-by-zero error, so it is necessary.

### B.1. Implementation Details: Baselines

#### B.1.1. DQN

We used the DQN (Mnih et al., 2013) implementation available in Stable Baselines. In the DQN algorithm, the optimal Q function is approximated by a neural network, $Q(s, a)$, trained using temporal-difference (TD) learning: $Q(s, a) \leftarrow Q(s, a) + \alpha \left( r + \gamma \max_{a'} Q(s', a') - Q(s, a) \right)$, where $\alpha$ is the learning rate. We set DQN with the following hyper-parameters for the learning rate of $5 \times 10^{-4}$, initial exploration rate $\varepsilon_{init} = 0.95$ decaying to $\varepsilon_{final} = 0.1$. A discount factor $\gamma = 0.99$, batch size of 64, target network update interval of 500 steps, one gradient step per update, and training frequency set to 1 (i.e., training occurs at every step).

#### B.1.2. PPO

We used the PPO implementation (Schulman et al., 2017b) available in Stable Baselines. In the PPO algorithm, the policy is updated via a clipped surrogate objective to ensure monotonic improvement without large destructive updates: $L^{CLIP}(\theta) = \hat{\mathbb{E}}_t[\min(r_t(\theta)\hat{A}_t, \text{clip}(r_t(\theta), 1 - \varepsilon, 1 + \varepsilon)\hat{A}_t)]$, where $r_t(\theta)$ is the probability ratio and $\hat{A}_t$ is the advantage estimate. We set PPO with the following hyper-parameters for the learning rate of $3 \times 10^{-4}$, horizon length $N_{steps} = 2048$, batch size of 64, and optimization epochs of 10. A discount factor $\gamma = 0.99$, GAE parameter $\lambda = 0.95$, clipping range $\varepsilon = 0.2$, entropy coefficient of 0.01, and value function coefficient of 1.

### B.1.3. TRPO

We used the TRPO (Schulman et al., 2017a) implementation available in sb3-contrib. In the TRPO algorithm, the policy is updated by maximizing a surrogate objective subject to a hard constraint on the KL divergence to ensure the new policy does not deviate excessively from the old one: $\max_\theta \hat{\mathbb{E}}_t \left[ \frac{\pi_\theta(a_t|s_t)}{\pi_{\theta_{old}}(a_t|s_t)} \hat{A}_t \right]$ subject to $\hat{\mathbb{E}}_t[\text{KL}[\pi_{\theta_{old}}(\cdot|s_t), \pi_\theta(\cdot|s_t)]] \leq \delta$. We set TRPO with the following hyper-parameters for the value function learning rate of $1 \times 10^{-3}$, horizon length $N_{steps} = 512$, batch size of 512 for value function updates, and 20 critic updates per policy step. A discount factor $\gamma = 0.99$, GAE parameter $\lambda = 0.98$, target KL divergence $\delta = 0.01$, conjugate gradient max steps of 15, and CG damping of $1 \times 10^{-3}$. We selected TRPO as a baseline because its trust-region constraint provides theoretically monotonic improvement guarantees and high stability in continuous control tasks, serving as a rigorous standard for comparison against newer, potentially less stable algorithms.

### B.2. Implementation Details: Reward shaping

We compared the performance of DQN, PPO, TRPO, and `ICR-RL` using the following reward-shaping functions.

#### B.2.1. ACROBOT-V1

We constructed a shaped reward for Acrobot by computing the vertical height of the end-effector (tip of the second link) based on the angular state:

$$r_{\text{shaped}} = -\cos(\theta_1) - \cos(\theta_1 + \theta_2).$$

This encourages the agent to raise the tip of the pendulum, promoting upright configurations. We assume both links have unit length.

#### B.2.2. MOUNTAINCAR-V0

We used a dense reward that combines progress and velocity:

$$r_{\text{shaped}} = \frac{\text{x} - x_{\min}}{-x_{min} + x_{max}} + c \cdot |\text{v}| - 1.$$

Where $x_{min} = -1.2$ and $x_{max} = 0.6$, and these define the environment borders, while $v$ is the velocity of the car. We use $c$ as a constant to encourage velocity and define $c = 10$. This formulation encourages forward motion and penalizes stagnation, helping the agent build momentum toward the goal.

#### B.2.3. CARTPOLE

We constructed a shaped reward based on normalized deviations from the center and upright position:

$$r_{\text{shaped}} = 2 - \frac{|\text{x}|}{x_{\max}} - \frac{|\theta|}{\theta_{\max}},$$

where $x_{\max} = 2.4$ and $\theta_{\max} = 12°$. This encourages the pole to remain near vertical and prevents the cart from moving outside the environment's bounds.

## C. Experiments: Additional Ablation Study

### C.1. `ICR-RL`: Number of Iterations

We conducted an experiment to test the sensitivity of `ICR-RL` to the number of FQI iterations. We set the number of FQI iterations to be equal to $K = \frac{\alpha}{1-\gamma}$, and study the performance as $\alpha$ modifies. We expect `ICR-RL` to converge in its performance for $\alpha \simeq 1$, as this is the number of iterations required for FQI to approximately converge. Surprisingly, we find that $\alpha = 0.1$ has the best performance on the MountainCar task (see Figure 8). This suggests that practitioners should consider using fewer iterations of `ICR-RL` than suggested by FQI theory.

### C.2. `ICR-RL`: Exploration Parameter

Modifying the $\varepsilon$ schedule, which includes the exploration parameter $\varepsilon$ and the $\varepsilon$-decay parameter $\lambda$, significantly affects how our method collects new episodes into the context. This is because an episode is added to the context only if its total

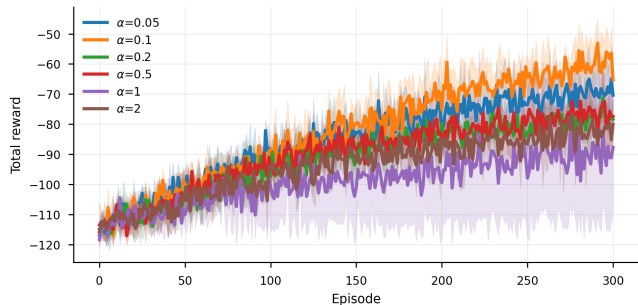

*Figure 8.* Performance of `ICR-RL` on MountainCar using different $\alpha$ values. The performance degrades significantly as $\alpha$ increases. This is in stark contrast to the FQI theory.

reward exceeds a threshold. Finding such episodes depends entirely on our method's ability to explore new state-action sequences that yield better results; therefore, the specific $\varepsilon$-schedule setting would likely affect our method's performance. We conducted ablation studies to assess the effect of the $\varepsilon$ schedule. This includes the initial $\varepsilon$ value $\varepsilon_0$ and the $\varepsilon$ decay parameter $\lambda$. The results show that the $\varepsilon$ schedule matters and leads to performance variability in the `ICR-RL` algorithm.

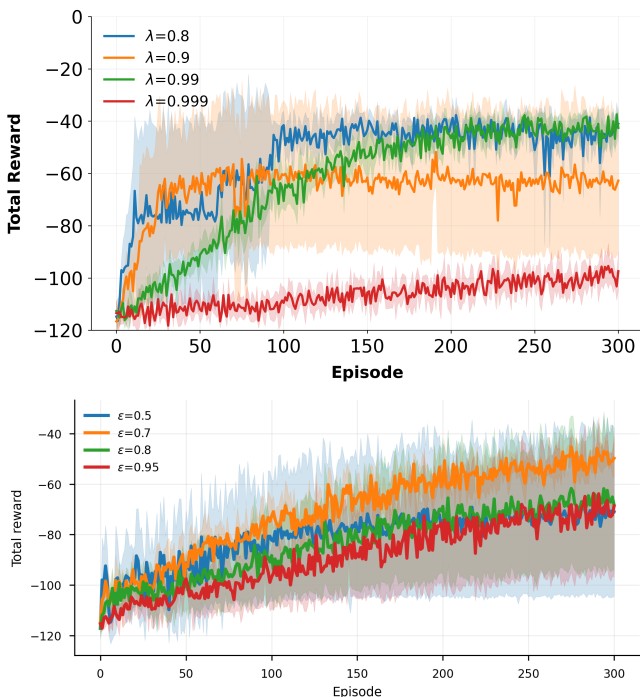

*Figure 9.* In the first plot, for the $\varepsilon$ decay, we considered values of 0.8, 0.9, 0.99, and 0.999 and found that 0.99 is the most stable (in green). In the second plot, we tested different values of the initial $\varepsilon_0$: 0.5, 0.7, 0.8, and 0.95, and observed that 0.7 achieved the best performance.

## D. Experiments: Additional Baselines

In addition to the experiments shown in the main paper, we added more baselines to show that our method works better than traditional methods with FQI such as K Nearest Neighbours, while also comparing the performance of TabPFN as the ICR backbone to different ICR foundation models as the ICR backbone such as TabICL (Qu et al., 2026) and TabFlex (Zeng et al., 2025). Our results show that our method significantly outperforms traditional methods, whereas TabICL outperforms TabPFN in one of the three environments. These experiments show the generality of our method and that different future work on ICR models may significantly improve our framework.

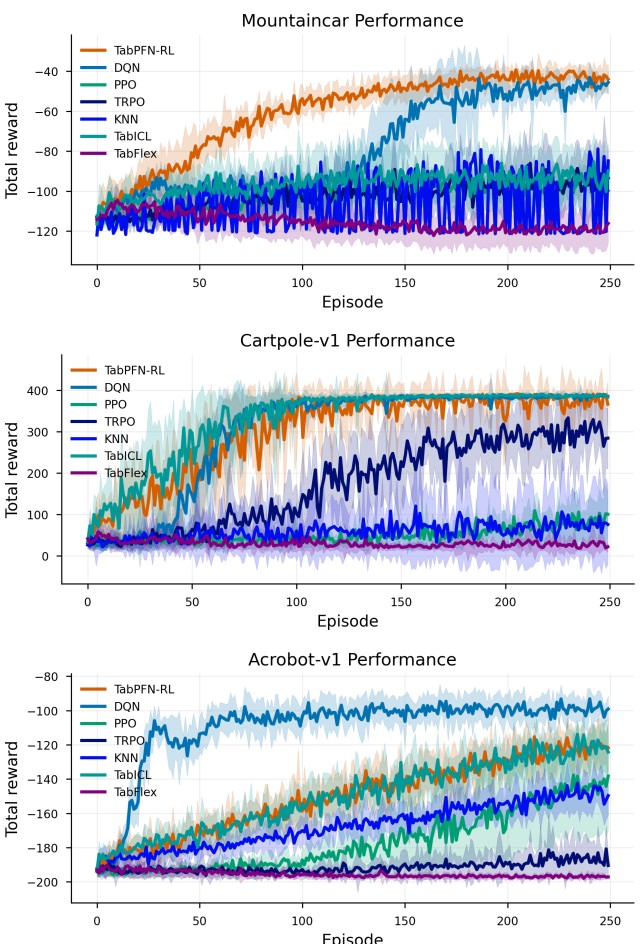

*Figure 10.* Comparison of TabPFN, DQN, PPO, TRPO, KNN, TabICL, and TabFlex in the ICR-RL framework on different environments.

