# OpenReview forum: "ICR-RL: Deep Reinforcement Learning via In-Context-Regression"
_ICML.cc/2026/Conference — ICML 2026 regular_

### Official Review · Reviewer_Wj4j · 2026-02-23

**Soundness:** 3
**Presentation:** 2
**Significance:** 3
**Originality:** 3
**Overall Recommendation:** 4
**Confidence:** 4

**Summary:**

The paper proposes a **training-free** reinforcement learning framework that leverages the in-context learning capabilities of pretrained regression transformers. Building on the classical Fitted Q-Iteration (FQI) formulation, the authors reduce the dynamic programming procedure of RL to a sequence of regression problems, thereby enabling the direct use of a regression transformer as a Q-function estimator. Concretely, the method iteratively relabels context trajectories with updated Q-value targets predicted by the transformer itself, allowing value iteration to be carried out purely through inference without parameter updates.

The overall idea is conceptually sound, as it composes well-established components (FQI and in-context regression) in a novel way, establishing an explicit connection between reinforcement learning and supervised regression that enables the use of general-purpose regression transformers. The authors empirically validate the approach on three classic control tasks, demonstrating competitive performance, and further discuss limitations as well as practical engineering techniques required to make the method effective.

**Compliance With Llm Reviewing Policy:**

Affirmed.

**Key Questions For Authors:**

**Q1.** As the policy improves, the Q-function should ideally be updated to reflect the latest policy. However, in Algorithm 1 it appears that ICR-FQI is executed only when a new episode is added to the context buffer and the buffer size is still below the context budget. Does this imply that once the context buffer is full, the Q-values are no longer refit? If so, how does the method continue improving the value estimates thereafter?   Could the authors clarify precisely when and how often the Q-values are recomputed with the transformer, especially after the context budget is reached?

**Limitations:**

Yes

**Strengths And Weaknesses:**

## Strengths

- The paper introduces a novel perspective on in-context reinforcement learning by reducing RL to regression via FQI, making it possible to directly use pretrained regression transformers for RL control without additional training.

- The core idea is conceptually well grounded, as it composes well-established components whose effectiveness are individually validated in prior work.

- The method achieves satisfactory performance on the evaluated classic-control benchmarks. It is particularly interesting that the approach works at all given the mismatch between the regression pretraining distribution and RL data.

- The paper provides a thorough empirical study of context-engineering strategies under fixed context budgets, which is a practically relevant contribution for in-context RL methods.

- The authors clearly acknowledge the limitations of the approach and outline reasonable future directions, especially regarding scaling and prior–data mismatch.


## Weaknesses

- The empirical evaluation is limited to a small set of simple classic-control tasks. While this is acceptable for a proof-of-concept, demonstrating effectiveness on more challenging domains would be important to assess the broader impact and generality of the approach.

- The method relies on reward shaping in these environments, which weakens the claim of general applicability to reinforcement learning settings.

- The paper would benefit from clearer writing and more precise wording in several sections to improve readability and technical clarity.

---

> ### Author Rebuttal · Authors · 2026-03-30
>
> We sincerely thank the reviewer for the detailed and constructive feedback, and for recognizing the motivation and potential impact of our work. We address the reviewer’s question and the identified limitations below.
>
> **W1:** We thank the reviewer for this important point. We agree that extending to high-dimensional settings (e.g., vision or robotics) is a natural next step. Our goal in this work is to isolate a core and surprising result: a pretrained in-context regression model, trained on i.i.d. tabular data, can already perform non-trivial control through value iteration, without any gradient updates or RL-specific training. We focus on classic control to demonstrate this clearly, without additional complexity from representation learning. Importantly, the framework itself is not restricted to low-dimensional inputs. It can be extended by combining it with pretrained encoders or by training in-context regression models on higher-dimensional data. We view this work as establishing the paradigm, with scaling to more complex domains as a clear and promising next step.
>
> **W2:** Our new additional experiments actually suggest that our method (ICR-RL) can perform well on sparse reward environments and is tied to the pretrained in-context regression backbone rather than to the ICR-RL framework itself. In particular, when replacing **TabPFN** with **TabICL-v2** [1] on CartPole without any reward shaping, we observed substantially stronger performance (2x) and a clear learning curve, with **TabICL-v2** [1] outperforming **TabPFN** by a large margin in this setting. This suggests that the sparse-reward limitation is not inherent to reducing RL to in-context regression, but depends on the prior induced by the backbone model. We will clarify this distinction more explicitly in the main text and present new results showing that our method also handles sparse rewards.
> **We present our additional results in ** https://anonymous.4open.science/r/unknown_project-6A74/
>
> **W3:** We appreciate this helpful feedback and agree that clarity can be further improved. In the main text, we will refine the presentation by clarifying the algorithmic flow. Our goal is to make the paper as accessible and clear as possible while preserving its technical depth.
>
> **Q1:** Regarding Q1, the reviewer raises an important point about the behavior of the ICR-RL algorithm. In Algorithm 1, we present what we refer to as a “stale” version of the algorithm, where updates cease once the context buffer is full. As the reviewer noted, this would indeed be problematic if left unaddressed. In practice, however, we explicitly handle this issue through the context engineering strategies described in the paper. These methods enable continued learning even after the context budget is reached. Specifically, context engineering dynamically removes less informative or redundant data points from the context, thereby freeing capacity for newly collected episodes. This allows the model to continuously refresh its context and recompute Q-value estimates, effectively maintaining an ongoing improvement process despite the fixed context size. We will revise the main text to make this mechanism more explicit and avoid ambiguity in the current presentation of Algorithm 1.
>
> [1] TabICLv2: A better, faster, scalable, and open tabular foundation model, 2026 https://arxiv.org/abs/2602.11139

---

> > ### Author Rebuttal · Reviewer_Wj4j · 2026-04-04
> >
> > I think my questions are much answered by the replies.
> > Given the scores I observed across the papers, I think my original evaluation is pretty reasonable for the this work.
> > It provides an innovative way to use transformers for in-context learning. Although the experiments have some limitations with it, I still keep an positive perspective to encourage this kind of work to appear.
> > I'll keep my score to be positive as a result.

---

### Official Review · Reviewer_y8Nz · 2026-03-04

**Soundness:** 4
**Presentation:** 4
**Significance:** 4
**Originality:** 3
**Overall Recommendation:** 4
**Confidence:** 3

**Summary:**

In most existing approaches, foundation models (FMs) are trained directly on a suite of control tasks. However, developing diverse RL environments and scaling training across them can be costly and complex. Against this backdrop, this paper proposes leveraging a pretrained model (TabPFN) that already possesses In-Context Regression (ICR) capabilities to carry out Bellman iteration and online control without parameter updates—using no gradients and pure forward inference only. In this way, it realizes a “reasoning-as-learning” deep RL framework, and achieves performance on classic control tasks that is close to DQN.

**Compliance With Llm Reviewing Policy:**

Affirmed.

**Key Questions For Authors:**

1. How does ICR-RL’s performance change in environments without reward shaping, i.e., using only the original sparse rewards?

2. Have you tried mitigating the signal-propagation issue under sparse rewards by modifying the target construction in ICR-FQI, and providing a fair comparison against DQN and offline RL baselines?

**Limitations:**

yes

**Strengths And Weaknesses:**

**Strengths**

1. ICR-RL uses ICR-FQI to recast value iteration as “in-context regression + relabeling,” with no backpropagation and no parameter updates throughout, making the framework more lightweight and the implementation path simpler.

2. It can continue learning under a fixed context budget. Even when the context window is full, performance can still improve, and a smaller context can achieve performance close to that of a larger context.

**Weaknesses**

1. The model’s inference complexity grows with the context length, which implies limitations in efficiency and scalability.

2. It is not well-suited to sparse-reward settings, and the experiments rely on reward shaping.

---

> ### Author Rebuttal · Authors · 2026-03-30
>
> We thank the reviewer for the thoughtful and detailed feedback. We also appreciate that the reviewer has acknowledged key contributions in our work, such as continual learning even when the context budget is depleted and the introduction of a gradient-free RL method without additional pretraining and reduced hyperparameters. We address the concerns raised by the reviewer below.
>
>
> **W1 Efficiency and scalability (context length growth):**
> We agree that the dependence of inference cost on context length is an important consideration. At the same time, one of the exciting aspects of ICR-RL is that this property is entirely inherited from the chosen backbone, rather than from the framework itself. This makes scalability a highly tractable and modular direction.
> To demonstrate this, we conducted additional experiments with **TabFlex [1]** , a linear-attention tabular model. **TabFlex [1]**  achieves a **~2.9× speedup** over **TabPFN**, illustrating that the framework can immediately benefit from more efficient architectures. While **TabFlex [1]** currently underperforms because it is primarily designed for classification, these results highlight a clear and promising direction: combining ICR-RL with regression-oriented linear-attention models can yield both strong performance and substantial efficiency gains.
>
>
> **Q1 and W2 Sparse Rewards**: Our additional experiments reveal that ICR-RL can perform well with sparse rewards and does not require reward shaping, but is governed by the choice of regression backbone. In particular, when replacing **TabPFN** with **TabICL-v2 [2]** and evaluating on CartPole without reward shaping or modifying hyperparameters, we observe a nearly 100% improvement over **TabPFN**, along with a clear, stable learning curve. (**See the results here**:  https://anonymous.4open.science/r/unknown_project-6A74/).
>
> This result is important because it shows that our method has strong generalization capacity; sparse-reward performance depends on how well the pretrained regression prior aligns with the task's structure. **TabICL-v2 [2]**, for example, is trained on synthetic datasets with tree-like structures, which are discrete and naturally align with sparse, branching reward dynamics. In contrast, **TabPFN’s** prior is less aligned with such a structure.
>
> More broadly, this highlights a key opportunity enabled by our framework:
> ICR-RL separates learning dynamics from model design, making performance directly improvable through better priors rather than more complex training procedures. This suggests a clear and scalable path forward, designing or pretraining regression models on more RL-like or structured data distributions could significantly improve performance in sparse and long-horizon settings.
>
> The ability to plug in different priors and observe large performance differences indicates that ICR-RL can serve as a flexible foundation for future work on data-driven RL priors.
>
> **Q2**: All baseline methods we compared used the same reward shaping scheme for fair comparison. Additionally, we conducted further experiments with other offline RL methods, specifically using KNN as the FQI backbone, and we show that these methods are unstable and significantly underperform compared with ICR-FQI. We share our results in the anonymous repository as reported above.
>
> [1] Zeng, Y., Dinh, T., Kang, W., & Mueller, A. C. Tabflex: Scaling tabular learning to millions with linear attention, 2025. URL https://arxiv. org/abs/2506.05584.
> [2] TabICLv2: A better, faster, scalable, and open tabular foundation model, 2026 https://arxiv.org/abs/2602.11139

---

> > ### Author Rebuttal · Reviewer_y8Nz · 2026-04-03
> >
> > I appreciate the authors' efforts in addressing my concerns.

---

### Official Review · Reviewer_Qosf · 2026-03-13

**Soundness:** 2
**Presentation:** 3
**Significance:** 2
**Originality:** 3
**Overall Recommendation:** 4
**Confidence:** 3

**Summary:**

This paper proposes ICR-RL, a framework that reduces reinforcement learning to in-context regression. Built on top of a specific pretrained regression model, TabPFN, In-Context Fitted Q-Iteration (ICR-FQI) iteratively updates value estimates within a context window without gradient-based fine-tuning. Experiments conducted on Gymnasium classic control tasks demonstrated that the proposed method can achieve performances competitive to classic RL methods like DQN, PPO and TRPO. In addition, this work introduces Native De-duplication (ND), a heuristic context engineering technique for continual learning with a fixed token budget by removing duplicate transitions from the context.

**Compliance With Llm Reviewing Policy:**

Affirmed.

**Final Justification:**

I appreciate the additional results provided by the authors, and my concerns have been fully resolved. Therefore, I raise my score from 3 to 4.

**Key Questions For Authors:**

1.	How to guarantee the framework can work well in such control tasks, which do not share the same data prior as the pretrained TabPFN?
2.	Only tested on low-dimensional tasks. How does the proposed method perform on high-dimensional tasks with visual features or complex robot state spaces?
3.	Does it mean that the proposed method can only work in the designed experiments with TabPFN? Can ICR-RL scale well when we replace TabPFN with another light-weight regression model?

**Limitations:**

1.	The current method is not compatible with high-dimensional observations.
2.	Heavily dependent on the TabPFN model. No evidence to scale to more general or similar models.
3.	High computational cost. The model is still bounded by the quadratic complexity of standard transformers.

**Actionable Items:**

1.	Consider replacing the standard transformer-based TabPFN with a linear attention mechanism (e.g, TabFlex [1]) to improve the runtime efficiency.
2.	Pretraining includes non-stationary data to better align the statistical properties of RL data.
3.	Test ICR-RL on sparse reward environments to help people better understand to what extent it will fail.
4.	Extend experiments to high-dimensional tasks to demonstrate the scalability of the proposed method.

[1] Zeng, Y., Dinh, T., Kang, W., & Mueller, A. C. Tabflex: Scaling tabular learning to millions with linear attention, 2025. URL https://arxiv. org/abs/2506.05584.

**Strengths And Weaknesses:**

**Strengths:**

1.	This work studies an alternative to reduce reinforcement learning into regression. The proposed framework demonstrates that a pretrained regression model can be directly applied to RL problems without gradient-based training. The advantage is that ICR-RL enables zero-shot generalization without requiring any RL-specific pretraining or fine-tuning.
2.	Since no fine-tuning is involved, this method reduces the need for hyperparameter searches in traditional RL methods.
3.	The introduction of context engineering techniques enables the model to perform well in small context sizes.

**Weaknesses:**

1.	The proposed method was heavily dependent on the pretrained regression model, TabPFN, showing low scalability. There exists a significant gap between the TabPFN data prior and the testing Gymnasium environments. TabPFN was trained on iid data, while RL data is usually non-stationary and correlated. How to guarantee the framework can work well in such control tasks? Moreover, the transformer-based TabPFN still requires high runtime costs even without gradients.
2.	The proposed method looks like a naïve combination of existing approaches, e.g, TabPFN and FQI, showing limited novelty in method design.
3.	The proposed method does not consistently outperform baselines in control tasks. In addition, the experiments indicate ICR-RL does not work well in sparse reward environments.

---

> ### Author Rebuttal · Authors · 2026-03-30
>
> **W1+AI1 Dependence on TabPFN:** Following this comment, we added results using other in-context tabular foundation models, such as **TabICL-v2 [1] and TabFlex [2]**. TabICL-v2 achieves performance comparable to **TabPFN** on CartPole and outperforms standard RL baselines on other environments, while **TabFlex [2]** provides about 2.9× faster inference. We present our additional results in https://anonymous.4open.science/r/unknown_project-6A74/
> These results strengthen the main point that the contribution is the reduction in ICR-FQI/ICR-RL, and shows that our method is general and not tied to a specific backbone.
>
> **i.i.d. mismatch:+AI2** Correlated and non-stationary data are inherent to RL, and many standard methods already rely on procedures to make training data more i.i.d.-like (e.g., DQN uses experience replay). Our framework follows a similar principle through the context: as more trajectories are accumulated, the dataset becomes more diverse and less temporally correlated, improving suitability for regression.
> To evaluate the effect of **non-stationary pre-training**, we fine-tuned **TabPFN** on trajectories from Cartpole (100 episodes), which improved performance and stability across 5 seeds, reaching the maximum reward with roughly 3% fewer episodes and lowering the std from 121 to 119. This preliminary positive signal indicates that, as suggested by the reviewer, training on non-stationary data has potential and can be further improved by using a different regression backbone trained on a different prior.
>
> While transformer inference has a runtime cost, the method is fully gradient-free, avoids repeated optimization, and can benefit from more efficient backbones. In particular, linearized-attention tabular models offer a clear path toward improved scalability.
> Overall, we position the paper as validating a new paradigm for RL, showing that pretrained in-context regressors can already solve nontrivial control tasks, with **TabICL-v2** and **TabFlex [2]** further demonstrating generality and scalability.
>
> **W2 Architectural novelty:** The novelty is not in introducing a new architectural module, but in identifying a **new use case and algorithmic role** for pretrained in-context regressors as the core engine for control in reinforcement learning. A model trained for tabular i.i.d. regression is used within an RL procedure with correlated, non-stationary, and bootstrapped targets, yet yields competitive control behavior. This demonstrates a non-trivial transfer, revealing a new capability and suggesting broader principles for repurposing foundation models.
>
> **W3 Support of sparse rewards:** We agree that ICR-RL does not consistently outperform all baselines and do not claim universal superiority. Our goal is to show that a **gradient-free, in-context regression framework** can achieve competitive control performance.
> We conducted additional experiments suggesting this is **not a limitation of ICR-RL itself** but of the backbone. While TabPFN underperforms in sparse-reward settings, replacing it with **TabICL-v2** on CartPole, **without reward shaping or hyperparameter changes**, yields a **significant improvement (AUC: 96.85 ± 37.66 vs 48.73 ± 13.92)**. This indicates performance depends strongly on the **inductive bias of the regression prior**.
>
> **Q1 Performance guarantee:** We do not claim a formal guarantee under prior mismatch; our claim is empirical. Despite the gap between **TabPFN’s** tabular prior pretraining and RL data, the framework performs competitively. Additional results with **TabICL-v2** and **TabFlex [2]** show that transfer depends on the backbone, reinforcing that the key idea is the **ICR-RL reduction**, while stronger priors can further improve performance.
>
> **Q2:** We agree that extending to high-dimensional settings (e.g., vision or robotics) is a natural next step. Our goal is to isolate a core result: a pretrained in-context regression model trained on i.i.d. tabular data can perform non-trivial control via in-context value iteration without gradient updates. The framework itself is not restricted to low dimensions and can be extended using pretrained encoders or higher-dimensional in-context models.
>
> **Q3:** Our additional experiments show that replacing **TabPFN** with **TabICL-v2** can improve performance out of the box in sparse-reward settings. This demonstrates the **generality and flexibility** of ICR-RL and highlights the importance of the regression prior.
>
>
>
> **AI3:** We added experiments showing how **TabPFN** performs **without reward shaping**, while **TabICL-v2 significantly outperforms it**, showing a strong learning curve while **nearly doubling the total reward**.  (results in https://anonymous.4open.science/r/unknown_project-6A74/).
>
> **AI4:** As noted in Q2, extending the method to vision encoders and complex state spaces is a promising next step and part of future work.

---

> > ### Author Rebuttal · Reviewer_Qosf · 2026-04-03
> >
> > Thanks for your efforts to resolve my concerns by providing additional results with TabICL-v2 and TabFlex and clarifications. I will raise my score from 3 to 4.

---

### Decision · Program_Chairs · 2026-04-30

**Decision:**

Accept (regular)

**Comment:**

This paper introduces ICR-RL, a framework that reduces reinforcement learning to a sequence of in-context regression problems. By utilizing pretrained regression foundation models like TabPFN, the authors enable value iteration and control through pure forward inference without the need for gradient-based fine-tuning or RL-specific training. Key strengths identified by reviewers include the novel perspective on using general-purpose regression models for RL and the method's ability to achieve competitive performance on classic control tasks with reduced hyperparameter tuning. However, critics pointed out significant weaknesses such as limited scalability, high computational costs due to transformer quadratic complexity, and poor performance in sparse-reward environments. The authors also introduced context engineering techniques to manage fixed token budgets, though the evaluation was largely restricted to low-dimensional benchmarks.

In the rebuttal, the authors addressed concerns about model dependence and scalability by providing additional results using alternative backbones, including TabICL-v2 and the linear-attention model TabFlex. They clarified that performance issues in sparse-reward settings were primarily due to TabPFN's specific inductive bias rather than a flaw in the ICR-RL framework itself. To resolve questions about non-stationary data, the authors argued that context accumulation naturally diversifies the data and presented preliminary fine-tuning results that improved stability. They also clarified the algorithmic flow, explaining how context engineering prevents Q-value updates from becoming stale once the buffer is full. While these responses satisfied the reviewers, some concerns remain about the framework's effectiveness in high-dimensional, complex environments such as robotics or vision-based tasks.

I am slightly leaning toward acceptance (weak accept); however, if there is no room in the program, I would not mind if my recommendation was bumped down.